# Effect of Pre-Processing Storage Condition of Cell Culture-Conditioned Medium on Extracellular Vesicles Derived from Human Umbilical Cord-Derived Mesenchymal Stromal Cells

**DOI:** 10.3390/ijms23147716

**Published:** 2022-07-13

**Authors:** Adrienne Wright, Orman L. Snyder, Lane K. Christenson, Hong He, Mark L. Weiss

**Affiliations:** 1Department of Anatomy and Physiology, Kansas State University, Manhattan, KS 66506, USA; adrwright@kbibiopharma.com (A.W.); ljsynder@ksu.edu (O.L.S.); hhe@vet.ksu.edu (H.H.); 2Department of Molecular and Integrative Physiology, University of Kansas Medical Center, Kansas City, KS 66160, USA; lchristenson@kumc.edu; 3Department of Anatomy and Physiology, Midwest Institute of Comparative Stem Cell Biotechnology, Kansas State University, Manhattan, KS 66506, USA

**Keywords:** exosomes, MSCs, extracellular vesicles, sample storage, transmission electron microscopy, nanomaterials, sarcoma cell proliferation

## Abstract

EVs can be isolated from a conditioned medium derived from mesenchymal stromal cells (MSCs), yet the effect of the pre-processing storage condition of the cell culture-conditioned medium prior to EV isolation is not well-understood. Since MSCs are already in clinical trials, the GMP-grade of the medium which is derived from their manufacturing might have the utility for preclinical testing, and perhaps, for clinical translation, so the impact of pre-processing storage condition on EV isolation is a barrier for utilization of this MSC manufacturing by-product. To address this problem, the effects of the pre-processing storage conditions on EV isolation, characterization, and function were assessed using a conditioned medium (CM) derived from human umbilical cord-derived MSCs (HUC-MSCs). Hypothesis: The comparison of three different pre-processing storage conditions of CM immediately processed for EV isolation would reveal differences in EVs, and thus, suggest an optimal pre-processing storage condition. The results showed that EVs derived from a CM stored at room temperature, 4 °C, −20 °C, and −80 °C for at least one week were not grossly different from EVs isolated from the CM immediately after collection. EVs derived from an in pre-processing −80 °C storage condition had a significantly reduced polydispersity index, and significantly enhanced dot blot staining, but their zeta potential, hydrodynamic size, morphology and size in transmission electron microscopy were not significantly different from EVs derived from the CM immediately processed for isolation. There was no impact of pre-processing storage condition on the proliferation of sarcoma cell lines exposed to EVs. These data suggest that the CM produced during GMP-manufacturing of MSCs for clinical applications might be stored at −80 °C prior to EV isolation, and this may enable production scale-up, and thus, and enable preclinical and clinical testing, and EV lot qualification.

## 1. Introduction

Mesenchymal stromal cells (MSCs) show promise as a cellular therapeutic, and they have been isolated from many tissues including bone marrow (BM), adipose tissue, and the placenta and umbilical cord (UC-MSCs) [1,2]. MSCs’ therapeutic effects have been attributed to paracrine signaling via secreted factors and extracellular vesicles (EVs), and by contact-mediated signaling [3,4,5,6,7]. EVs are lipid-bilayer membrane-enclosed particles that are divided into three major subpopulations based on size and origin: microvesicles (MVs, 50–1000 nm), apoptotic bodies (50–5000 nm), and exosomes (30–150 nm) [4,8,9,10,11]. EVs are involved in both physiological and pathological signaling [12,13,14,15,16], and can be isolated from many biological fluids including cell culture-conditioned media (CM). We previously showed that HUC-MSCs-derived EVs traffic to sarcoma tumor models and thus, may aid in diagnostic imaging [17]. Following up that work, we observed that the pre-processing storage conditions, and EV isolation methods had impacts on EV trafficking (Cromer, Weiss, and Basel, unpublished observations), other laboratories have also noted that pre-processing and isolation methods affect EV production from a CM [6,18,19,20].

The downstream application of and laboratory-to-laboratory comparison of EVs is challenging due to the lack of universally standardized methods for pre-processing conditions, EV isolation, and EV characterization. Thus, while standardized, reproducible methods for characterizing EVs have been proposed by the International Society for Extracellular Vesicle (ISEV) in position papers [21,22,23], MSC-EVs are moving into clinical testing with open questions regarding the optimal pre-processing storage conditions and EV isolation method [24].

The impact of pre-processing storage conditions on the CM and other biological fluids on EV isolation/characterization is unclear and is reviewed briefly below. With regard to CM-derived EVs, there is no consensus with some laboratories showing that pre-processing storage conditions are not critical considerations prior to EV isolation [25,26,27], and others showing that freeze/thaw damage occurs, and that cryopreservation and proper storage temperatures are important [28,29,30,31,32]. For example, Lee et al. reported that EVs isolated from HEK 293 cell CM and stored at temperatures of −70 °C, 4 °C or room temperature for 10 days had protein and RNA amounts reduced after room temperature storage compared to storage at −70 °C and 4 °C [33]. Furthermore, storage at 4 °C and room temperature resulted in loss of the exosome surface marker CD63 expression while storage at room temperature caused the loss of Hsp70 expression [33]. Lee et al. concluded that −70 °C was the ideal storage temperature [33]. In contrast, Cheng et al. demonstrated that EVs isolated from HEK 293T cell CM had the highest number of particles and levels of EV characterization proteins (ALIX, Hsp70, and TSG101) when stored at 4 °C compared to EVs stored at 37 °C and 60 °C or subjected to freeze/thaw cycles [34]. Using nanoparticle tracking analysis (NTA), Sokolova et al. demonstrated that EVs isolated from HEK 293T CM had a greater reduction in size when stored at 37 °C compared to 4 °C [35]. In a patent owned by Capricor Therapeutics, the number of EVs isolated from the cardiosphere-derived cells CM remained stable when stored for one week at 4 °C, −20 °C, and −80 °C, the miRNA concentration decreased significantly in EVs stored at 4 °C and −20 °C yet remain unchanged in those stored at −80 °C [36]. In contrast to CM-derived EVs, EVs derived from cerebrospinal fluid of glioblastoma patients were found to be stable at room temperature for seven days, as well as after a single cycle of freeze-thawing, while a second freeze-thaw cycle resulted in significant alterations of EV number, morphology, and RNA content or miRNA levels [37]. Lorincz et al. demonstrated that storage of EVs isolated from neutrophilic granulocytes at 4 °C and −20 °C resulted in a decrease in both the number of EVs and the antimicrobial effect of EVs, while storage at −80 °C had no effect on either the number or antimicrobial effect [38]. Milk-derived EVs are impacted by pre-processing conditions prior to isolation, and the optimal conditions differ across species [39]. In sum, as reviewed above and noted previously by other researchers [30], there is no consensus on the optimal pre-processing storage conditions. While the ISEV has provided recommendations regarding characterization of EVs [21,22,23], the pre-processing storage of CM and other biological fluids such as plasma, milk, or spinal fluid has not been standardized, and no consensus exists.

As schematically shown in Figure 1, here, three pre-processing storage conditions for HUC-MSC-derived CMs were compared to CM samples processed immediately for EV isolation for effects on EV isolation and characterization including biophysical and functional outcomes. Pre-processing storage at room temperature, 4 °C, and −20 °C resulted in partially degraded EVs isolated from HUC-MSC CM, and thus, research findings using these less optimal storage conditions cannot be completely disregarded. In contrast, pre-processing storage in −80 °C resulted in consistent batches of EVs as indicated by polydispersity index, protein staining, and transmission electron microscopy. Thus, pre-processing storage at −80 °C would be recommended prior to EVs isolation from UC-MSC CM.

## 2. Results

### 2.1. EV Analysis by Nanoparticle Tracking Analysis (NTA)

NTA analysis of the EV population for each storage condition is shown in Table 1. NTA sizes ranged from 85 nm to 106 nm, with immediate isolation condition tending to be the smallest size (85.08 ± 21.7 nm) and frozen −20 °C tending to be the largest (105.42 ± 18.6 nm). No differences were detected in the size data between EVs isolated immediately or from the pre-processing storage conditions (*p* = 0.45). Based upon the effect size observed here, power analysis revealed that for a power of 0.8 and an alpha of 0.05, a sample size of 20 would be required to detect significant differences between storage conditions assuming normal and unimodal distributions via a two-tailed analysis.

The EV particle counts by the CM pre-processing storage condition as measured by NTA are shown in Figure 2. No differences in EV total particle count were detected among samples isolated immediately or from pre-processing storage conditions. Total particle counts ranged between 6.99 × 10^10^ ± 3.12 × 10^10^–1.28 × 10^11^ ± 8.30 × 10^10^. This indicates that the pre-processing storage condition did not significantly impact the number of EVs isolated from HUC-MSC CM samples. Based upon the effect size observed here, power analysis revealed that for a power of 0.8 and an alpha of 0.05, a sample size of 36 would be required to detect significant differences.

Using NTA particle count, EVs released per cell were estimated by averaging all pre-processing storage condition per cell line (Table 2). No significant difference in EV release per cell was detected across the five cell lines (*p* = 0.08) indicating that the five cell lines used produced a similar number of EVs in CM during the conditioning period.

The particles per cell was averaged by the CM pre-processing storage condition and is shown in Figure 3A. No significant differences were detected among experimental storage conditions (*p* = 0.263). Similar to the particle range observed by the individual cell line, the range by CM pre-processing storage condition was 1.73 × 10^4^ ± 7.67 × 10^3^ to 3.15 × 10^4^ ± 1.70 × 10^4^ particles/cell. Based upon the effect size observed here, power analysis predicted that a sample size of 18 would be required to detect significant differences among cell lines at a power level of 0.8 and alpha of 0.05. While the effect of HUC-MSC cell passage number (Figure 3B) or HUC-MSC population doubling time (Figure 3C) on the number of EV particles isolated was not be statistically evaluated since just five cell lines were sampled. No gross trends were observed.

### 2.2. EV Characterization by Dynamic Light Scattering (DLS)

EVs were characterized using DLS to determine the polydispersity index (PDI), zeta surface potential, and hydrodynamic size. The CM that was stored pre-processing at −80 °C was significantly different than the CM immediately processed for PDI (Figure 4A, *p* = 0.012), and there was a trend for the immediately processed EVs to be more dispersed (e.g., higher PDI) than the other pre-processing storage conditions. As shown in Figure 4B, the zeta potential of EVs was found to range from −7.73 ± 3.76 mV to −12.40 ± 2.50 mV. No differences were detected in the zeta potential between the pre-processing storage conditions (*p* = 0.143), and there was a trend for EVs from CM stored at all pre-processing storage conditions to be lower than EVs from CM that was isolated immediately. Based upon the observed effect size, power analysis revealed that a sample size of 12 would be required to detect significant differences at a power of 0.8 and an alpha of 0.05. As shown in Figure 4C, EVs isolated from CM had a hydrodynamic size of 165.64 ± 42.73 nm to 410.67 ± 262.85 nm. The group differences did not reach significance (*p* = 0.052), and there was a trend for EVs particles in the immediately processed group to be larger than those stored at the −80 °C or room temperature prior to EV isolation (see Figure 4C). Note in Figure 4C, both room temperature and −80 °C pre-processing storage tended to have a smaller range than other groups. This narrower range in hydrodynamic size in the −80 °C pre-processing storage group fits with the significantly reduced PDI seen in Figure 4A. Taken together the DLS data suggests that −80 °C storage may produce a more homogenous population of EVs and smaller EVs compared to immediate processing.

### 2.3. EV Analysis by Transmission Electron Microscopy (TEM)

Using TEM, EVs were detected in all CM storage conditions. For each storage condition, 10 to 29 (average 22.3 ± 5.77) EVs were measured per independent sample (Figure 5A). In all conditions, EVs appear to be roughly spherical and ranged in diameter from 67.9 to 95.0 nm. The morphology of EVs in the immediately processed and −80 °C storage condition displays a prominent black ring indicating a bilayer structure whereas the other groups do not have as distinctive of a black ring (also described as a doughnut shape in the EV literature) with a slightly less dense core. As shown in Figure 5B, significant size differences were noted between EVs isolated immediately and those isolated after storage at −20 °C, as well as between room temperature storage and −20 °C storage condition groups. Similar to DLS data, EVs in the immediately processed group display a broader size range compared to other experimental groups. The TEM data suggests that storage of CM at −80 °C may produce a more homogenous population of particle size compared to immediate processing.

### 2.4. EV Characterization by Dot Blots

To assess protein staining, scorers rated the intensity of dot blots according to the scoring standard provided in Figure 6A. Following reconstitution of lyophilized samples, EVs were probed for expression of tetraspanins CD9, CD63, and CD81, as well as heat shock protein 70 (Hsp70) (Figure 6B). B-actin was used as a protein loading control and its expression was not scored. All storage conditions produced EV samples that had positive (defined as weak or strong staining) staining for at least three of the four characteristic markers but tended to vary in intensity. As shown in Figure 6C, significant differences were observed in the dot blot score of EVs from the immediate group compared to EVs isolated from CM stored at −80 °C (*p* = 0.005). The median score for the immediate isolation group was 3.7 while the median score for the −80 °C storage condition was 7.7. This result suggests that EVs isolated from the CM stored at −80 °C had higher intensity of staining for characteristic EV markers compared to EVs isolated immediately from the CM. Specifically, the biggest staining difference was seen for Hsp70, as represented in Figure 6B. EVs isolated from CM stored at −80 °C had a significantly higher average score for expression of Hsp70 compared to EVs isolated immediately from CM (1.4 vs. 0.5, respectively; *p* = 0.023).

### 2.5. Analysis of EV-Associated miRNAs

To assess whether CM storage temperature has an effect on miRNA yield from EVs, we analyzed the yield of miRNA for five independent samples per storage condition. As shown in Figure 7, no significant differences were found (*p* = 0.07) but there was a trend for RNA from CM refrigerated at 4 °C (18.3 µg/mL) less than other storage conditions. Yields from immediate (30.5 µg/mL), room temperature (25.7 µg/mL), −20 °C (28.1 µg/mL), and −80 °C (24.8 µg/mL) were consistent.

RT-PCR was performed and the results of four representative miRNAs are shown in Figure 8A (all RT-PCR results are found in Supplemental Data Figure 1). Based on the PCR results, EVs contain miRNA for all miRNAs tested (miR-4466, miR-1273e, miR-4792, and miR-127, miR-658, miR-1246, miR-3665, miR-6089, and miR-7641). Figure 8B shows a heat map of the miRNA results by storage condition with 5 being the maximum value and 0 being the minimum. Results for miR-1273e were consistent across all storage conditions. All samples except two had a positive result for miR-127. EVs from the CM stored at −20 °C displayed maximum scores (i.e., expression by all samples) for miR-127, miR-1246, and miR-3665 indicating that storage at −20 °C did not affect expression of these miRs. EVs from the CM stored at −80 °C displayed maximum scores for miR-127 and miR-1246 indicating that storage at −80 °C did not affect expression of these miRs. Results for miR-4792, miR-6089, and miR-7641 were the most inconsistent among all EV samples. In Figure 8C, mean scores were compared by storage condition for all miRNAs tested. No significant differences were found (*p* = 0.846). There was a trend for immediately-isolated EVs to score lower (5.2) compared to room temperature (6.0), 4 °C (6.2), −20 °C (7.0), and −80 °C (6.4).

**Figure 8 ijms-23-07716-f008:**
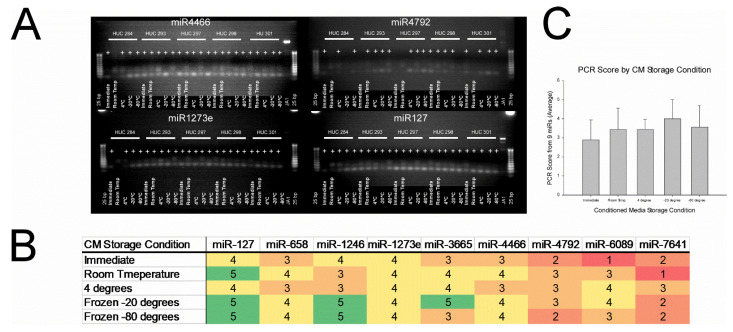
Effect of storage condition on miRNA expression. No significant differences were detected (*p* = 0.846) between storage conditions. Analysis of extracellular vesicle (EV)-associated microRNAs (miRNAs) by real-time polymerase chain reaction (RT-PCR) for EVs isolated from human umbilical cord-derived mesenchymal stromal cells (HUC-MSCs) condition media (CM) immediately or stored at room temperature, 4 °C, −20 °C, or −80 °C. (**A**) RT-PCR analysis of MSC-associated EV miRNA. miR primers were designed using the NCBI genebank data for human miR-127, miR-658, miR-1246, miR-1273e, miR-3665, miR-4466, miR-4792, miR-6089, and miR-7641 (see Table 3). There is evidence that EVs contain miRNA for all primers tested. (**B**) Heat map of PCR scores by CM storage condition. Individual samples were scored a 1 for positive expression or 0 for negative expression. Scores were added per miR for each storage condition for a maximum score of 5. Scores can range from 0 to 5 with green representing high scores and red depicting low scores. EVs in the −20 °C had the most maximum scores followed by −80 °C and then room temperature. Neither immediate nor 4 °C had a maximum score. miR-1273e was consistent among all storage conditions. (**C**) Mean PCR scores for EVs by CM storage condition. There was a trend for immediately-isolated EVs to have a lower score (5.2) compared to room temperature (6.0), 4 °C (6.2), −20 °C (7.0), and −80 °C (6.4). Data are presented as mean ± standard deviation, *n* = 5.

### 2.6. Effect of EV Storage Conditions on Sarcoma Cell Proliferation

One critical element to characterizing EVs is assaying their functional output. Previous work has shown that EVs can impact cellular proliferation in a dose and tissue of origin manner [40]. We hypothesized that EVs might affect sarcoma cell growth in a dose-response fashion, and that EV function might be impacted by the CM storage condition prior to isolation. We tested seven different sarcoma cell lines including Ewing’s sarcoma (1 line), rhabdomyosarcoma (2 lines), and osteosarcoma (4 line) after exposure to 5 different doses of EVs ranging from 0 through 1 × 10^8^ EVs per well, and EVs immediately isolated versus EVs isolated after the CM storage at −80 °C for up to one month prior to processing. ANOVA found no significant main effects of storage conditions, sarcoma cell line, or EV dose, or interactions. These results are summarized in Figure 9.

## 3. Discussion

Here, the effect of the pre-processing storage condition of the CM prior to EV isolation was analyzed. Four new findings encapsulate this work. First, the CM storage conditions did not affect the number of EVs that were isolated from the five conditions tested. The five HUC-MSC lines produced fairly consistent numbers of EVs in the range of 1.0–2.0 × 10^4^ particles per cell over the 24-h conditioning period. As expected with a small sample size, regression analysis did not detect any significant correlations between EV yield per MSC and a cell’s passage or population doubling time, and no gross trends were observed. Second, PDI results revealed that the storage of the CM at −80 °C produced a statistically different, and more homogenous population of particles compared to the CM which was immediately used for EV isolation, but this storage condition did not affect the zeta potential or the hydrodynamic size, but a trend was observed. Third, TEM revealed that EVs isolated from the CM stored at −80 °C exhibited similar doughnut shape morphology to immediately isolated EVs. In contrast, the EVs detected in samples from the CM stored at room temperature, 4 °C, and −20 °C did not show a clear doughnut shape, alluding to possible damage or degradation. Fourth, EVs from the CM stored at −80 °C prior to processing to isolate EVs displayed enhanced staining for CD9, CD63, CD81, and Hsp70 in protein blots compared to the other storage conditions. Again, suggesting improved preservation of the proteins in EVs by the ultralow storage temperature prior to EV isolation. These findings were used to estimate an appropriate sample size required to answer related research questions in follow-up work. They suggest that 30 samples would need to be evaluated to find significant differences between storage conditions based upon the standardized effect sizes observed. Taken together, these results suggest that pre-processing storage of HUC-MSC derived CM prior to EV isolation is an important consideration when considering EV manufacturing scale up and clinical testing. These results also highlight the need to identify a reliable functional assay that can be correlated with or predict outcomes, since the proliferation of sarcoma cells did not differentiation EVs from different pre-processing storage conditions. These results suggest that the pre-processing storage of HUC-MSC-derived CM at −80 °C for up to one month was optimal. These findings may apply to CMs stored for longer times or for CMs derived from other cell types or for biological fluids, but additional work would be needed for confirmation.

In EV literature, there is no standard method for isolation, but ultracentrifugation and SEC are two of the most common methods employed. When selecting an isolation method, available equipment, sample volume, and intended downstream use must be considered. Here, a combination of ultrafiltration and SEC was utilized. SEC is advantages compared to other methods, like ultracentrifugation, because it reduces protein contamination [41,42,43,44,45]. In our hands, SEC reduces protein concentration of EV samples by two orders of magnitude compared to UC (Abello, Snyder, Cromer, and Weiss, unreported observations). Additionally, SEC can be scaled up to accommodate large input volumes and requires no special equipment aside from a standard benchtop centrifuge. The downside is, to our knowledge, there are no GMP-qualified SEC supplies. Ultrafiltration allows for large volume samples to be concentrated and filtered, making SEC separation efficient. Here, ultrafiltration reduced the CM sample volume by ~100× prior to SEC. We investigated the impact of pre-processing storage conditions using the SEC EV isolation method, and we contend that it is likely similar results would be obtained should alternative methods for EV isolation, such as UC, be employed. However, to investigate the question of the interaction of pre-processing storage conditions and the EV isolation method is beyond the scope of the present report, and additional work would be needed to confirm or deny this speculation.

There is no consensus regarding the effects of the CM pre-processing storage condition from HUC-MSCs on the resulting EVs after isolation. Immediate isolation of EVs from the CM is the most common and was used as the standard for comparison here. Pre-processing conditions of room temperature, 4 °C, −20 °C, and −80 °C were selected for testing based on the existing literature, since we sought to test commonly used pre-processing storage conditions. In work by Zhou et al. and Romanov et al., CM storage temperatures 4 °C, −20 °C, and −80 °C were tested [31,46]. Romanov et al. also tested pre-processing storage at −196 °C but, with large CM volumes resulting from GMP-manufacturing, we consider this pre-processing storage condition as not feasible, and thus, it was not included here [31]. We assumed that room temperature storage of CM would result in the degradation of EVs, and thus, this condition was considered a negative control. In an unexpected result, HUC-MSC derived CM stored at room temperature for one week did not cause frank differences in the EVs isolated. This finding was confirmed by NTA, DLS, TEM, PCR, and dot blot analysis. This finding suggests stability of subpopulations of the vesicles isolated by SEC. Observation of the greater variability in a number of biophysical measures from EV derived from immediately processed CM, we believe that larger-size vesicles are lost over the one week of storage due to instability or loss due to freeze/thaw.

NTA was used to estimate the concentration of particles and size distribution of EVs. NTA found sizes ranging from 85 nm to 106 nm among CM storage conditions, similar to previous reports of EVs from BM-MSCs [6], and HUC-MSC derived EVs [17]. The NTA estimates of the size of the EV population were in the desired range (30–150 nm) for EVs of interest (i.e., exosomes) for each storage condition. The results indicate that the CM pre-processing storage does not impact the NTA based size estimates. Although a balanced experimental design was used to control for cell line-to-cell line variations, some experimental groups did show a high variance (e.g., immediate group had standard deviation of 25.5% of the sample mean) indicating that the data were dispersed relative to the mean. Other experimental groups had a lower variance (e.g., 4 °C pre-processing storage resulted in EVs with a standard deviation of 9.2% of the sample mean), indicating that the data was more clustered around the mean. Based upon the standardized effect size observed, a sample size of 20 would be required to detect significant differences. However, this estimate must be taken cautiously since it is based upon homogeneity of variance across the groups, which was not observed here.

NTA-based particle counts ranged from 6.99 × 10^10^ ± 3.12 × 10^10^–1.28 × 10^11^ ± 8.30 × 10^10^ particles per mL. Similar particle counts were found among different pre-processing storage conditions, indicating that the CM storage prior to EV isolation did not cause significant particle loss. Likewise, Zhou et al. also found that the storage of urine at −80 °C allowed for a comparable number of exosomes to be recovered compared to fresh urine [46]. In addition, Romanov et al. found that storage at 4 °C for 1 week did not cause degradation of microvesicles isolated from the CM [31]. In contrast, Romanov et al. reported that freeze/thaw cycles, regardless of storage temperature, caused significant degradation to microvesicles. One possible explanation for this finding would be that since exosomes are the smallest of the vesicles, and thus, contain a smaller amount of liquid inside, their membrane is less susceptible to damage from freeze/thaw cycles. In summary, the present work and that by others [37,46,47] demonstrate the stability of exosomes after freeze/ thaw cycles. Microvesicles and apoptotic bodies are larger than exosomes and, as a consequence, contain more liquid. Higher aqueous content may make these vesicles more susceptible to damage from freeze/thaw regardless of the temperature, as demonstrated by Romanov et al. [31].

The largest source of experimental variability in both size and particle count estimates came from the NTA measurements themselves. High variability makes it more difficult to detect a true difference when there is one. Here, NTA samples were analyzed using 5 × 60 s videos to produce an average measurement. Parsons et al. reported that increasing video replicates to 25 × 60 s reduced overall variance and increased precision of concentration estimates for particles between 50–120 nm from biological fluids [48]. Increasing the number of replicate videos captured used to calculate the sample concentration estimate may provide a more representative mean for samples and reduce experimental variation, thus making it easier to detect true differences in experimental groups without the need for large sample sizes. In other work, we observed that by using 1 × 10^8^ particles per mL, i.e., the high end of recommended particle counts for NTA measurement suggested by Malvern, or by using a blue laser rather than the green laser used here, we observed reduced variation in NTA particle count measurements (Snyder et al., in preparation).

The EVs released per MSC were estimated using the previous cell passage’s PDT, passage number, time in culture prior to removal of serum-containing medium, and the NTA concentration data. For this calculation two assumptions were made. First, MSCs would continue to grow at the same rate exhibited in their prior passage. Second, expansion would stop after the removal of the serum-containing medium. These assumptions are based upon our experience with these particular cell lines. First, HUC-MSCs isolation and culture conditions have been optimized so that they display consistent growth patterns [49]. Second, serum or serum alternatives are required for MSC expansion [50,51]. Particles per cell were compared by both the pre-processing storage condition and the individual cell line. No differences or obvious trends were noted. MSCs released an average of 2.19 × 10^4^ ± 8.5 × 10^3^ EVs per cell which is similar to previous reports [6,52]. In contrast, Crain et al. reported an average of 5.78 × 10^3^ ± 3.3 × 10^3^ EVs per MSC from canine UC-MSCs [53]. This disparity is likely due to species differences but may also be due to differences in EV isolation protocol, since UC is less efficient than SEC. While our protocol utilized SEC isolation, Crain et al. utilized density gradient ultracentrifugation [53]. Applying our EV isolation protocol to canine MSCs would help elucidate if this disparity is largely from species differences or EV isolation protocol differences. MSCs used here were healthy (>95% viability), of similar passage (P4–P6), and grown in the same culture conditions. We offer no explanation for the line-to-line differences other than biological variability. EV yield and characterization differences among cell lines would be a topic in future work.

Regression analysis did not detect any significant relationship between EV yield per MSC and a cell’s passage or population doubling time. This is due to the small sample size (*n* = 5) and also the few passages tested here (P4–P6). A trend for cell passage to be negatively correlated with EVs released per MSC is suggested. Thus, as a cell ages, perhaps fewer EVs are released. More work will be needed to confirm this observation. However, this is expected since aging cells are associated with cell senescence [6,54,55,56,57]. A second trend was noted when comparing EV yield per MSC by a cell’s PDT. Cells with a larger PDT (i.e., cells growing more slowly) tended to release more EVs per cell. Again, a larger sample will be required to confirm this finding.

DLS was used to characterize the PDI, zeta surface potential, and hydrodynamic size of the EVs. There were indications that EVs isolated immediately to be different than EVs in the −80 °C group in all DLS measurements. First, a significant difference in PDI was observed. Mean PDI values ranged from 0.336 to 0.635, similar to previously reported values [17]. PDI describes the size distribution in a sample [58]. As discussed by Danaei et al., PDI is a dimensionless measure and values greater than 0.7 indicate that the sample has a wide size distribution and is not considered to be ideal for analysis by DLS. Although all pre-processing storage conditions yielded EVs with PDIs less than 0.7, the immediately processed EV isolation group had a PDI of 0.635 compared to 0.336 for EVs from the −80 °C group. This suggests that storage of the CM at −80 °C produces a more uniformly-sized EV sample. This may occur due to freeze/thaw cycle damaging or deleting larger vesicles in the population, as previously discussed. Samples with a PDI larger than 0.7 are considered not ideal for DLS analysis. Immediately-isolated samples exhibited the highest PDI value (0.635) which is nearing the threshold. This presents a problem as DLS is regarded as one of the standard methods for EV characterization.

Zeta potential provided by DLS measures surface charge, colloidal stability, and integrity [52,59]. EVs carry a net negative surface charge under physiological conditions [59]. EVs in all experimental groups displayed a negative zeta potential, similar to previous reports [17,52,60,61]. Measured zeta potentials ranged from −7.73 ± 3.76 to −12.40 ± 2.50 mV. Although no significant differences in pre-processing storage conditions were detected, the immediately processed EV isolation group has a −7.73 ± 3.76 mV zeta potential. This value could be due to the presence of charged molecules that increase the ionic strength and may lead to low dispersion stability and compromised biological function [52,62]. Even so, the zeta potential of the EV samples was not as negative as others have reported [17,59]. As discussed by Midekessa et al., zeta potential can be significantly impacted by the buffer used for suspension, presence of salts and detergents, and the pH of the sample [59]. Here, EVs were suspended in degassed, DPBS without calcium and magnesium. No detergent or additional salts were added. The pH of the samples was not recorded, but were stored in pH 7.4 buffered solution, which was constant for all pre-processing storage conditions. From this data, the trend was for CM immediately used for EV isolation to yield EV population that would be less stable, based upon zeta potential, than EVs from the other storage conditions. This observation is supported by the significant change in PDI, and other trends reported.

Hydrodynamic size estimates measured via DLS were larger than sizes reported by NTA and TEM. The median size of EVs ranged between 165.64 ± 42.73 nm and 410.67 ± 262.85 nm, with no significant differences detected between pre-processing storage conditions. The largest median value was observed in the CM immediately processed EV isolation group (410.67 nm), suggesting the presence of larger EVs (i.e., microvesicles and apoptotic bodies), or agglomeration of vesicles. In contrast, the smallest median was observed in the −80 °C experimental group (165.64 nm) suggesting either a loss of the large EVs population, or reduced agglomeration. This finding supports PDI differences discussed above. The larger range noted in the immediate group and the narrow range in the −80 °C group supports the PDI differences observed. Taken together, this data indicates that pre-processing storage of CM at −80 °C produces a more homogeneous sample.

Using TEM analysis, EVs were detected in all pre-processing storage conditions. EVs have a spherical appearance and ranged in diameter from 67.9 to 95.0 nm. TEM size estimates were similar to both NTA size estimates reported here and previously reported size estimates [17]. In TEM micrographs, EVs in the immediately processed CM and in the pre-processed storage at −80 °C groups display a prominent black ring (doughnut shape) around the less electron dense center of the EV, indicative of the negative staining procedure. Despite the absence of the doughnut shape, EVs were detected in the CM stored at room temperature, 4 °C, and −20 °C demonstrating that none of the CM storage conditions tested here caused apparent degradation of EVs. Similar to DLS data, a broad size range was observed in the EVs isolated immediately after the CM collection, and the presence of larger EVs fits with DLS PDI and size measures. EVs from the CM which was stored at 4 °C, −20 °C, and −80 °C prior to EV isolation exhibited a narrower size range in TEM, with fewer large vesicles. In TEM, agglomeration was not observed, but we did occasionally observe EVs in close proximity. These findings further support our hypothesis that larger vesicles are lost due to freeze/thaw. Taken together, because of the narrower size range, and morphology in TEM that is most similar to the EVs isolated immediately from CM, storage at −80 °C appears to be the optimal CM storage temperature.

Protein staining to identify exosomes should include C membrane tetraspanins proteins CD9, CD63, and CD81, as well as endosomal markers such as Lamp2, Alix, Tsg101, Hsp70, etc. [21,22,23]. Ideally, these protein markers can be specifically identified by western blot to confirm protein size. One limitation of our work was that we used protein dot blotting to detect CD9, CD63, CD81 and Hsp70, rather than western blotting, followed by the scoring of the protein staining intensity by blinded investigators. Our reason for taking this direction was two-fold. First, the SEC isolation method yielded low protein concentrations (average 5.84 µg/mL). Thus, to load a western blot requires 2–10 µg protein per lane, thus a single western blot would be the majority of the sample aliquot. Second, positive staining was obtained by blotting 0.75 µg of protein. It is desirable to consume as little as possible of the EV sample for characterization, and this goal was met by dot blot. We obtained fairly consistent protein blot staining by first lyophilizing the EVs samples, resuspending them at a 5× higher concentration, and loading the PVDF membrane with 5–20 μL per dot. All pre-processing storage conditions produced blots that stained for at least three of the four markers tested here, but the results varied in expression intensity. After protein staining intensity was scored and the scores averaged, significant differences were observed between EVs isolated immediately compared and those from CM stored −80 °C prior to EV isolation. On a scale of 0–8, EVs isolated from CM immediately had a median score of 3.7 compared to a median of 7.7 for EVs in the −80 °C pre-processing storage condition. The difference in scores is due in large part to the difference in Hsp70 expression between the two groups. EVs in the −80 °C pre-processing storage condition the Hsp70 score was 1.4 (a score of 1.0 denotes weak, positive expression), while the staining intensity of EVs derived from CM immediately processed was 0.5. This indicates that storage of the CM at −80 °C allows EVs retained expression of Hsp70. Similarly, Lee et al. found that the storage of isolated EVs from HEK 293 CM at room temperature caused the loss of Hsp70 expression [33]. Higher protein staining overall of EV markers, and the higher staining of Hsp70, indicates that storage of CM at −80 °C is the ideal storage condition.

One critical missing piece to assessing the effects of CM storage conditions was an EV functional assay. Owning to the well-documented effects that MSC-derived EVs have on sarcoma cells in vitro and in vivo, we evaluated the effect of MSC-derived EVs, the CM storage condition prior to EV isolation, and EV dose on seven different sarcoma cell lines. It was hypothesized that MSC-derived EVs change the proliferation rate of sarcoma cells in vitro. “Doses” of 0 (negative control), 5 × 10^5^, 1 × 10^6^, 1 × 10^7^, and 1 × 10^8^ EVs per well were tested and no dose response relationship was observed. Second, seven different human sarcoma cell lines including Ewing’s sarcoma (1 line), rhabdomyosarcoma (2 lines), and osteosarcoma (4 different lines) were tested. No effects of EV dose, CM storage condition prior to isolation, or sarcoma cell type was observed. These negative results suggest that MSC-derived EVs from naïve cells have little effect on sarcoma cell proliferation. In future testing, we will evaluate with MSC-derived EVs change their phenotype to a cancer promoting one after the MSCs are co-cultured with sarcoma cells, as previously suggested [63,64]. Without a clear impact of EVs on sarcoma cell proliferation, we can’t make a strong conclusion whether −80 °C CM storage prior to EV isolation is different functionally from EVs isolated immediately after conditioning. This is a limitation of our work, and the field in general. Have a widely accepted functional assay would greatly enable laboratory to laboratory comparisons and allow replication via standardization of “dose”.

In summary, this work investigates the method of CM storage for downstream EV isolation based upon the hypothesis that CM used for isolation of EVs, like other cells and cell-derived products, may be impacted by storage condition and temperature. This report found that although all experimental storage conditions produced EVs, storage at −80 °C is the ideal storage temperature for HUC-MSC CM. A comparable number of EVs were recovered from CM stored at −80 °C compared to fresh CM. In addition, compared to fresh CM EVs, −80 °C EVs were of comparable size, had a more homogeneous population, more negative surface potential, shared a similar morphology with a distinctive bilayer structure, and had an overall higher protein staining of characteristic EV markers (CD9, CD63, CD81, and Hsp70). While significant findings are reported here, further work remains to test whether these similar characteristics translate to functionality of EVs. The knowledge gap in the storage of CM and other biological fluids prior to EV isolation should be closed in order to maximize utility for clinical and diagnostic translation.

## 4. Materials and Methods

### 4.1. Preparation of Conditioned Media from Human Umbilical Cord-Derived Mesenchymal Stromal Cells

Human umbilical cord-derived mesenchymal stromal cells (HUC-MSCs) were isolated, culture-expanded, characterized, and cryopreserved using a previously described protocol [49]. The research protocol was reviewed and approved by the Kansas State University human subject research committee and was considered exempted from human subjects research under exemption 4. From a cryobank of 57 HUC-MSC cell lines, 18 lines were selected randomly and subjected to full MSC characterization in accordance with the International Society of Cellular Therapy (ISCT) recommendations [65]. All lines demonstrated (1) plastic adherence and self-renewal via tissue culture expansion and clonal expansion capability, (2) tri-lineage differentiation capacity via qualitative differentiation assays to bone, cartilage, and fat lineages, and (3) positive surface marker expression (defined as >95% positive over isotype control) of CD73, CD90, and CD105 alongside negative surface marker expression (defined as <2% positive over isotype control) of CD34 and CD45. For the work described herein, five cryopreserved MSC lines were selected randomly from the 57 MSC lines for use and three of which had been fully characterized previously as MSCs. There were no frank differences observed between these five MSC lines in EV production.

The HUC-MSC lines (P3–P5) were thawed and plated individually at a density of 1 × 10^4^ cells/cm^2^ on tissue culture-treated T-150 flask (Corning, Cat. No. 430825) and allowed to recover from cryostorage for one passage before use in experiments. The cells were maintained in a culture and passed as previously described [49]. For quality control purposes, only cell lines with >95% viability during the passage prior to EV isolation were used for CM production (all five lines passed this criterion). As shown in the experimental schematic (Figure 1), following passage, HUC-MSCs were plated at a density of 1 × 10^4^ cells per cm^2^ in five T-150 flasks and incubated at 37 °C, 90% humidity, and 5% CO_2_ in a HeraCell 150i incubator in a growth medium containing platelet lysate [49]. Once cells reached 60–70% confluence (approximately 48 h), the medium was removed and replaced with 30 mL low glucose Dulbecco’s Modified Eagle Medium (DMEM, Gibco, Cat. No. 11885) without platelet lysate. After 24 h, the medium was considered conditioned (i.e., CM), and it was collected and transferred to a sterile 50 mL centrifuge tube. The collection tubes were pre-labeled with experimental conditions (immediate, room temperature, 4 °C, −20 °C, and −80 °C) and the order was randomized and blinded prior to the CM collection. The CM in the immediate processing group was collected and processed for EV isolation within 24 h. The CM in the room temperature, 4 °C, and −20 °C groups were stored for 7 days at their designated temperature and were then processed for EV isolation within 24 h. Due to the COVID-19 pandemic, core facilities were forced to close and the CM samples in the −80 °C group were stored up to 1 month before being processed for EV isolation. The experimental design was balanced to ensure that there was an equal number of observations for all possible level combinations thus, the CM from all lines were collected for every storage temperature. This was done to control for possible line-to-line variations in EV production.

### 4.2. Preparation of Size Exclusion Chromatography (SEC) Column

A slurry of Sepharose CL-2B (GE Healthcare, Cat. No. 65099-79-8) was prepared and degassed using sterile Dulbecco’s phosphate-buffered saline (DPBS, Gibco, Cat. No. 14-190-250) according to the manufacturer’s instructions. A 20 mL syringe (EXELINT, Cat. No. 26280) was stuffed with cotton wool and the column was poured with 20 mL of Sepharose CL-2B slurry. After settling overnight in a cold room, the column height was 7.5 cm and the column diameter was 2.21 cm. Several volumes of DPBS were rinsed through the column to ensure consistent packing. The void volume and separation properties were determined using blue dextran solution (Sigma-Aldrich, St. Louis, MO, USA, Cat. No. D5751) and bovine serum albumin (BSA, Sigma-Aldrich, Cat. No. A3912-500G), respectively, prior to EV isolation.

### 4.3. EV Isolation by a Combination of Ultrafiltration and Size-Exclusion Chromatography

EVs were isolated using a combination of ultrafiltration and SEC similar to Benedikter et al. with some modifications [66]. The CM was centrifuged at 3200× *g* for 30 min at 4 °C (Eppendorf 5810R using a swing bucket rotor A-4-62, Cat. No. FL08517291) to pellet cells and cellular debris. The supernatant was collected and filtered through a 0.22 µm syringe filter (Fisherbrand, Cat. No. 09-720-004). The CM was transferred to an Amicon Ultra-15 filter unit with Ultracel-100 membrane (MWCO = 100 kDa, Merck Millipore, Cat. No. UFC910024) and centrifuged at 3200× *g* until the retained sample volume was approximately 300 µL. This represented a 1:100 concentration factor approximately. The retentate was collected in a sterile microcentrifuge tube. The membrane was subsequently rinsed with 200 µL DPBS to collect any adherent EVs, and this was added to the sample tube.

The filtered sample (~500 µL) was layered onto the SEC column and eluted with sterile, degassed DPBS. Following the void volume, 27 fractions of 250 µL were collected. Protein analysis was performed on the fractions using a NanoDrop 8000 spectrophotometer (Thermo Scientific, Waltham, MA, USA). Following analysis, fractions were pooled, divided into 1 mL aliquots in polypropylene microcentrifuge tubes, and stored at −80 °C until EV characterization.

### 4.4. Lyophilization of EVs

To freeze-dry samples, frozen 1 mL aliquots were removed from the −80 °C. Immediately a small hole was made in the top of the microcentrifuge tube using a sterile 20 G × 1-inch needle (Becton Dickinson, Cat. No. 305175) and the lip of the lid was trimmed using scissors so that it could fit inside of a sterile 15 mL polypropylene centrifuge tube (Nunc, Cat. No. 339650). This was done so that the samples could be freeze-dried at a 45–60° angle and to catch any particles that spilled over from the hole in the tube. The lids of the 15 mL centrifuge tube were replaced with lids that had 5 holes previously made using a 20 g needle. The samples were loaded into a TF-10A 1.2-L vacuum freeze dryer (TEFIC BIOTECH CO., LIMITED, Xi’an, China), which has a condenser capable of reaching ≤56 °C and a vacuum of <10 Pa. Samples were lyophilized in batches of 5–6 samples overnight for 16–18 h. Following lyophilization, samples were stored at room temperature in the 15 mL centrifuge tube with original lids and sealed with parafilm to protect from moisture. Immediately prior to use, an aliquot of lyophilized EVs was rehydrated with sterile UltraPure distilled water (Invitrogen, Cat. No. 10977) to 20% of their original volume and vortexed. Following lyophilization and rehydration, the protein concentration was measured. Lyophilized/reconstituted EV samples were used for transmission electron microscopy and dot blot analysis.

### 4.5. Nanoparticle Tracking Analysis

Nanoparticle tracking analysis (NTA) was used to assess the EV population size distribution and concentration using a NanoSight LM-10 (Malvern Pananalytical Ltd., Malvern, UK) using a previously described protocol [67]. Briefly, measurements were made at a constant temperature of 25 °C ± 1, using blue 405 nm laser, camera type scientific CMOS, camera level 13, and detection threshold 3. NanoSight software (NTA 3.3) analyzed 60-s videos with 5 repetitions per sample. Five independent samples from each storage temperature were analyzed, and 5 technical replicates were averaged for comparison via repeated measures analysis of variance (ANOVA). Prior to use, the calibration was checked using 50 nm and 100 nm standards (Malvern Pananalytical Ltd., Cat. Nos., NTA4087 and NTA4088, respectively). Sterile DPBS was used as the negative control, and sample dilutions ranged from undiluted to 1:100 in sterile DPBS to ensure that sample measurements fell in the NTA optimal range of 1 × 10^8^ to 1 × 10^9^ particles/mL (30–50 particles/frame).

The NTA particle concentration data was used to estimate EVs released per cell. Population doubling time (PDT) was calculated for each cell line using the previous passage data according to the equation below.
$$Population\;Doubling\;Time=\frac{Duration\;of\;Culture\;\left(hours\right)\times \log\left(2\right)}{\log\;\left(final\;cell\;count\right)-\log\;\left(initial\;cell\;count\right)}$$

An estimate of the number of MSCs was calculated using the PDT and the number of hours that the cells have been in the culture (data shown in Table 4). The assumption was made that when cell culture media were replaced with a serum-free medium, expansion stopped. The EVs released per cell were estimated as the total particles in the sample divided by the estimate of cells in the culture. Five independent samples from each storage temperature were analyzed. The five independent measurements for particles per cell were averaged by both storage condition and by individual cell line and compared using one-way repeated measures ANOVA. The effect of MSC expansion (e.g., passage number and population doubling time) on EVs released per cell was evaluated. These results were plotted and a simple linear model was derived using regression.

### 4.6. Dynamic Light Scattering

Dynamic light scattering (DLS), zeta potential, and polydispersity index (PDI) were used to assess the hydrodynamic size distribution, surface charge properties, integrity, and stability of the EVs, as previously described [17]. Briefly, measurements were made using the Zetasizer Nano ZS (Malvern Pananalytical., Malvern, UK) [17]. Instrument settings were 10 runs of 10 s with 3 repetitions per sample. Five independent samples from each storage temperature were analyzed. Measurements were made with technical triplicates and averaged for comparison via one-way repeated measures ANOVA.

### 4.7. Transmission Electron Microscopy (TEM)

Electron microscopy was used to visualize EV morphology and to generate a size estimate of the population. Lyophilized samples were rehydrated using sterile DPBS. The samples were prepared for negative staining, experimental conditions were masked, and the samples were blindly analyzed at the University of Kansas Medical Center’s TEM facility, as previously described [40,41]. An area of approximately 69 µm^2^ was sampled and a minimum of 20 micrographs were collected. The size of the EVs was estimated by measuring data from two MSC lines that were chosen at random in all storage conditions. A minimum of 20 micrographs were captured per storage temperature and all EVs depicted were measured. From these data, sizes were averaged by storage condition to generate a size measurement to be compared using Kruskal–Wallis one-way ANOVA on ranks.

### 4.8. Protein Measurement of EV Samples

The protein concentration of the individual fractions from the SEC was determined using the absorbance at 280 nm on a NanoDrop 8000 spectrophotometer (Thermo Scientific, Waltham, MA, USA) immediately following elution from the column. DPBS was used as the blank for background subtraction. This was done to detect any protein contamination in the samples and ensure proper separation of samples.

Once the fractions were combined, the protein concentration of the 1 mL EV sample fractions were determined using a Pierce micro BCA protein assay kit (Thermo Scientific, Cat. No. 23235) according to the manufacturer’s instructions. Samples were plated in technical triplicates in a 96 well plate (Corning, Cat. No. 3370). The absorbance was read at 562 nm using a SpectraMax i3x plate reader (Molecular Devices, San Jose, CA, USA). A protein standard curve was generated using BSA standards supplied with the kit and by averaging technical triplicates. DPBS was used as a blank for background subtraction. Samples were plated in triplicate and the average was used to determine the protein content of the EVs.

### 4.9. Dot Blot

To perform dot blots, samples were reconstituted in 200 µL distilled water after lyophilization. The protein content in the samples was determined using the absorbance at 280 nm on the NanoDrop spectrophotometer with distilled water as the blank for background subtraction. The protein content for each sample was calculated as the average of three technical replicates.

To perform dot blots, 0.75 µg of protein was loaded onto an activated PVDF membrane (MilliporeSigma, Burlington, MA, USA, Cat. No. IPVH0010). After protein dots were completely dried, the membrane was blocked with 5% non-fat dried milk solution. The membrane was probed with mouse primary antibodies: anti-human CD9 (1:1000, Ts9, Invitrogen, Cat. No. 10626D), anti-human CD63 (1:500, Ts63, Invitrogen, Cat. No. 10628D), anti-human CD81 (1:500, M38, Invitrogen, Cat. No. 10630D), anti-human Hsp70 (1:200, 3A3, Santa Cruz Biotechnology, Cat. No. sc-32239), and anti-human B-actin (1:2000, AC-74, Sigma-Aldrich, Cat. No. A2228) overnight at 4 °C with gentle rocking. Water and lysed MSCs (i.e., whole cell lysate) served as negative and positive controls, respectively. Following incubation with primary antibodies, membranes were washed three times using 1% tris-buffered saline with 0.1% Tween-20 detergent and blotted with secondary antibody HRP-conjugated goat anti-mouse IgG (1:2000, Poly4053, BioLegend, Cat. No. 405306) for 1 h at room temperature with gentle rocking. After washing, chemiluminescence detection reactions were performed using SuperSignal West Femto substrate (Thermo Scientific, Cat. No. 34095) according to the manufacturer’s instructions. Images were captured using a Kodak Image Station 4000 after 2 min of exposure.

To score dot blot staining, 3 independent scorers were provided numbered dot blot strips, such that the experimental conditions were masked as to protein blotted and experimental group information, and a scoring standard (shown in Figure 6A). Scorers were instructed to judge the intensity of the dot only, not the size of the dot itself. Dots were scored on an ordinal (linear) integer scale as strong (2 points), weak (1 point), or negative (0 points). Expression of B-actin was not scored since it was used for a positive control. For each blotted protein, the score was averaged from three scorers. Averaged scores were summed for the four markers per MSC line and storage condition giving a total score for comparison. The maximum score possible was 8.0 (strong positive expression for all four markers). The total score for each storage condition and cell line from the five independent lines was analyzed using repeated measured ANOVA on ranks and pairwise comparisons were made using Dunn’s method.

### 4.10. MicroRNA (miRNA) Isolation and Reverse Transcriptase-Polymerase Chain Reaction (RT-PCR)

MicroRNA was isolated and reverse transcribed (RT) into a cDNA template using RNAzol RT (Sigma-Aldrich, Cat. No. R4533) according to manufacturer’s instructions. Briefly, RNAzol RT was added to EV samples. Samples were covered, shaken vigorously, and then left at room temperature for 5–15 min. The mixture was centrifuged at 12,000× *g* for 15 min at 4–28 °C. The supernatant was carefully removed and transferred to a clean microcentrifuge tube, leaving a small layer above the pellet of DNA and proteins. To precipitate mRNA, 75% ethanol *v*/*v* was added to the supernatant and let stand at room temperature for 10 min. The sample was centrifuged to pellet mRNA. The supernatant was transferred to a clean microcentrifuge tube. To pellet miRNA, 100% isopropanol *v*/*v* was added to the supernatant, let stand at 4 °C for 30 min, and then centrifuged. Both mRNA and miRNA were washed using 75% ethanol followed by 70% isopropanol and centrifuged at 15,000× *g* for 1–3 min at room temperature. Alcohol was discarded and RNA pellets were reconstituted in RNase-free water. To check for protein contamination, a 260/280 ratio was obtained for each sample using a NanoDrop 8000 spectrophotometer (Thermo Scientific, Waltham, MA, USA).

The Poly(A) tail reaction and cDNA synthesis was performed using the MystiCq microRNA cDNA Synthesis Mix (Sigma-Aldrich, Cat. No. MIRRT) according to manufacturer’s instructions. Briefly, the miRNA sample was combined with Poly(A) polymerase, buffer, and water. The sample was sealed, vortexed, centrifuged, and incubated at 37 °C for 60 min. The resulting reaction mixture was combined with ReadyScript Reverse Transcriptase and MystiCq microRNA cDNA reaction mix then sealed, vortexed, centrifuged, and incubated for 20 min at 42 °C. Polymerase chain reaction (PCR) was performed using a BioRad iCycler: the initial denaturation at 98 °C for 30 s, 35 cycles of [98 °C for 15 s, 51 °C for 30 s, and 72 °C for 5 s], and the final extension at 72 °C for 10 min. After the PCR, the products were resolved on a 2% agarose gel with a 25 bp DNA ladder and imaged using ethidium bromide. The RNAs and the MystiCq Universal PCR primer (Sigma-Aldrich, Cat. No., MIRUP) are provided in Table 3. miRNAs for analysis were chosen from the EVmiRNA database. Using the database, MSCs were selected and the database returned 9 miRNAs known to be expressed in EVs from MSCs [http://bioinfo.life.hust.edu.cn/EVmiRNA, filter = mesenchymal stem cell, accessed on 6 March 2020]. Primers were designed using the NCBI database and the sequences, along with accession number, are listed in Table 3.

Samples were scored based on the presence or absence of a band of correct size (as determined by the primer length and addition of PolyA tail. A sample was scored a 1 for positive signal and a 0 for negative signal. Scores were added for each independent cell line per storage condition for a maximum score of 5 and a minimum score of 1. Scores were compared by storage condition using one-way ANOVA.

### 4.11. Effect of EVs on Sarcoma Cell Line Proliferation

The effect of different “doses” of EVs were tested (0, 5 × 10^5^, 1 × 10^6^, 1 × 10^7^, 1 × 10^8^ particles/mL) on seven different sarcoma cell lines (osteosarcoma (MG-63, SJSA-1, Saos-2, 143B), rhabdomyosarcoma (RD, SJCRH30), and Ewing’s sarcoma (A-673), all from ATCC) on cell proliferation. Sarcoma cell lines were taken from cryogenic storage, plated into a 6-well tissue culture plate, and grown to confluency over 4 days. The cells were plated, passed and counted, as described above. Next, the cells were then seeded in triplicates into a 12-well tray and grown for two days. After two days, EVs were added and left to proliferate for an additional two days before passing and measuring the live cell number via AO/PI. EV samples isolated immediately from CM were compared to those processed after CM storage at −80 °C storage prior to isolation were tested for cell proliferation via cell count and AO/PI staining. ANOVA was used to evaluate the effects of sarcoma cell line, storage conditions, and dose.

### 4.12. Statistics

After checking to verify the ANOVA assumptions were met, ANOVA was used to evaluate the main effects and interactions. After finding significant ANOVA main effects or interactions, post hoc testing of pre-planned comparisons was performed using either Bonferroni correction or Dunn’s method. These data are presented in figures or in text as mean ± one standard deviation, unless stated otherwise. For pairwise comparisons, statistical assumptions were confirmed, then the data was analyzed using Student’s *t*-test. If ANOVA assumptions were not met, Kruskal–Wallis analysis of variance on ranks was used for statistical testing. Data from nonparametric tests were presented using a box and whisker plot depicting the median, 25th, and 75th percentile in the box, and whiskers showing the 10th and 90th percentile. Any potential outliers were depicted as open circles. Hypothesis testing was two-tailed, and a *p* < 0.05 was considered “significant.” In some cases, a power analysis was conducted to determine the sample size needed to detect significant differences using a desired power of 0.8 and an alpha of 0.05. For regression analysis, either SigmaPlot 14.5 or the statistical package R (version 4.1.1) was used. The entire dataset for each experiment, including potential outliers, was used throughout. Most statistical analyses and graphs were generated using SigmaPlot (version 12.5–14.5, Systat Software, Inc., Palo Alto, CA, USA). Graphs were exported as enhanced metafiles (EMF), were edited using Canvas X (version 15.5, build 1770 or version 19.0, build 333, Canvas GFX, Inc., Boston, MA, USA), and saved in uncompressed TIFF format.

## 5. Conclusions

The goal here was to identify an optimal storage condition for conditioned media prior to EV isolation. Storage at −80 significantly reduced the polydispersity index, suggesting that a more uniform size population is found after one freeze/thaw cycle. Based upon dot blot staining, we conclude that storage at −80 °C prior to EV isolation appears to provide the optimal condition for extracellular vesicles.

## Figures and Tables

**Figure 1 ijms-23-07716-f001:**
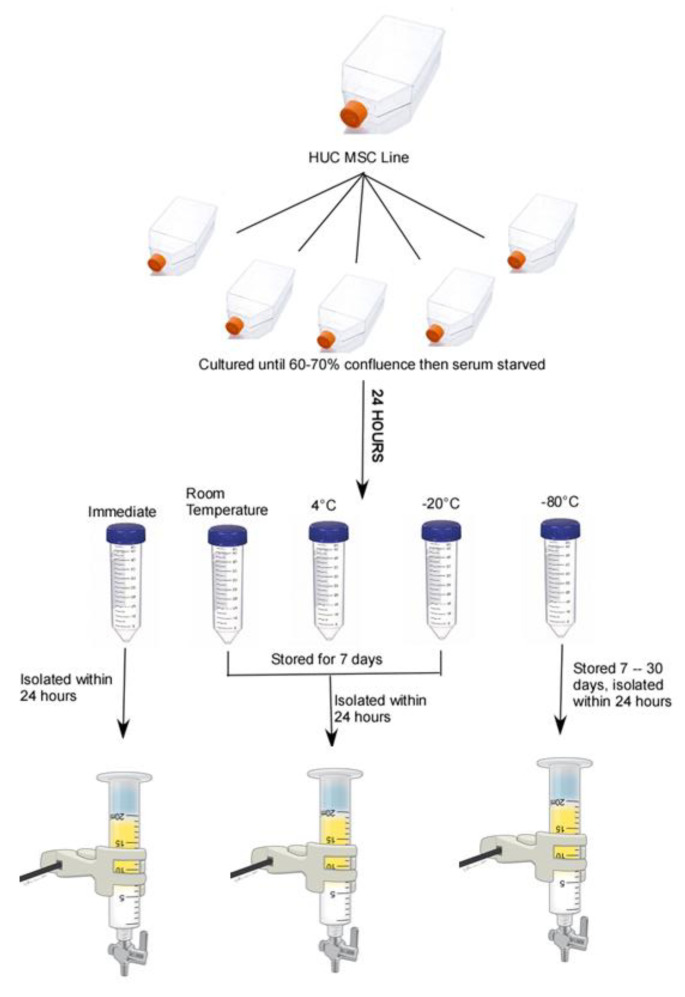
A schematic of the balanced experimental design showing the major steps involved for the isolation of extracellular vesicles from the human umbilical cord-derived mesenchymal stromal cell (HUC-MSC) culture-conditioned media. Following cryopreservation, HUC-MSCs were allowed one passage of recovery before being passaged and split into five T-150 tissue culture-treated flasks and incubated at 37 °C, 90% humidity, and 5% CO_2_ until 60–70% confluence was reached. The medium was removed and replaced with a serum-free medium. After 24 h, conditioned media were collected in a sterile 50 mL centrifuge tube and randomly assigned one of the five storage conditions (immediate, room temperature, 4 °C, −20 °C, and −80 °C). Samples in the immediate group were isolated within 24 h by a combination of ultrafiltration and size-exclusion chromatography. Samples in room temperature, 4 °C, and −20 °C were stored for seven days at the assigned temperature and isolated within 24 h. Samples in the −80 °C group were stored up to one month before EV isolation.

**Figure 2 ijms-23-07716-f002:**
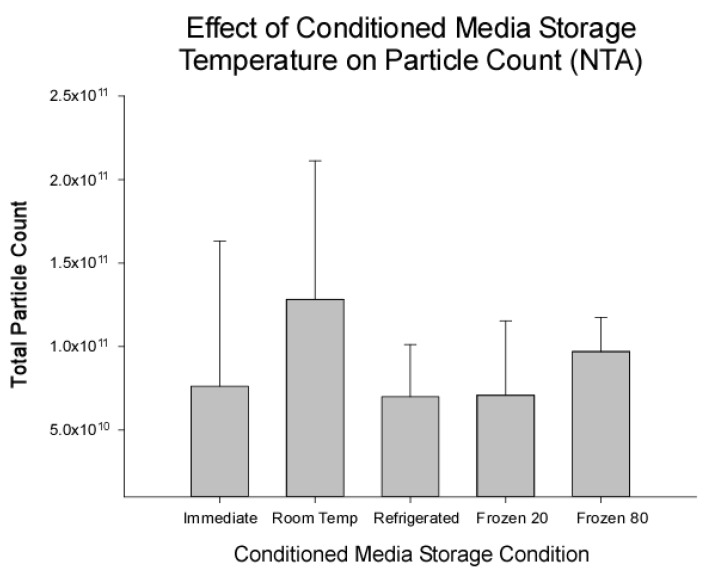
Storage temperature of conditioned media (CM) did not affect particle count measured by Nanoparticle Tracking Analysis (NTA). Particle counts ranged from 6.99 × 10^10^ ± 3.12 × 10^10^ to 1.28 × 10^11^ ± 8.30 × 10^10^. No significant differences in particle count were detected (*p* = 0.332) indicating that CM storage temperature prior to isolation did not significantly impact the number of particles isolated from the sample. Data are presented as mean ± standard deviation (*N* = 5).

**Figure 3 ijms-23-07716-f003:**
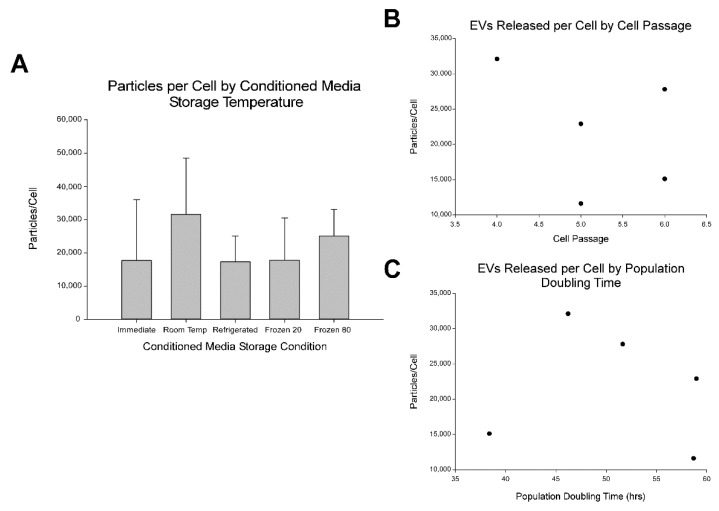
Pre-processing storage condition did not affect the number of EV particles recovered from human umbilical cord-derived mesenchymal stroma cells (HUC-MSCs) conditioned medium. (**A**) Number of EVs released per HUC-MSC by conditioned media (CM) pre-processing storage condition (immediate, room temperature, 4 °C, −20 °C, and −80 °C). EV particles released per MSC ranged from 1.73 × 10^4^ ± 7.67 × 10^3^ to 3.15 × 10^4^ ± 1.70 × 10^4^. No significant differences were detected. Thus, CM pre-processing storage conditions had no significant impact on number of EV particles recovered. Data are presented as mean ± standard deviation and is representative of five independent cell line measurements. (**B**) Analysis of EVs particles released per MSC versus cell passage. No trends were observed. (**C**) Analysis of EVs released per cell by population doubling time. No trends were observed.

**Figure 4 ijms-23-07716-f004:**
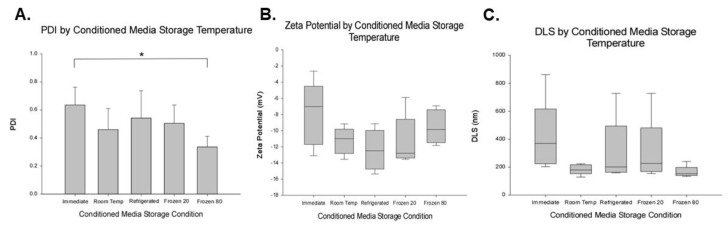
Pre-processing storage condition effects on dynamic light scatters data. The polydispersity index (PDI) in (**A**), the zeta potential in (**B**), and dynamic light scattering (DLS)- based EV size in panel (**C**). Note in (**A**), there was a significant decrease in PDI, indicating that pre-processing storage at −80 °C had a less complex solution in terms of particle size distribution compared to the solution obtained by immediate isolation of EVs. In both (**B**) and (**C**), no significant effects of pre-processing storage condition on zeta potential (**B**) or DLS-based particle size (**C**). However, clear trends were observed in DLS data (**A**–**C**), suggesting that the immediately processing to isolate EVs was a more diverse and complex solution compared to that obtained after pre-processing storage at −80 °C. In figure A, ANOVA main effect significant, followed by post hoc pre-planned comparisons. In panel A, averages (means) + standard deviation are shown. In panels (**B**) and (**C**), box and whisker plots show median, first, and third quartile in box and 10th and 90th percentile in whiskers. Asterisk depicts significant difference, *p*-value < 0.05.

**Figure 5 ijms-23-07716-f005:**
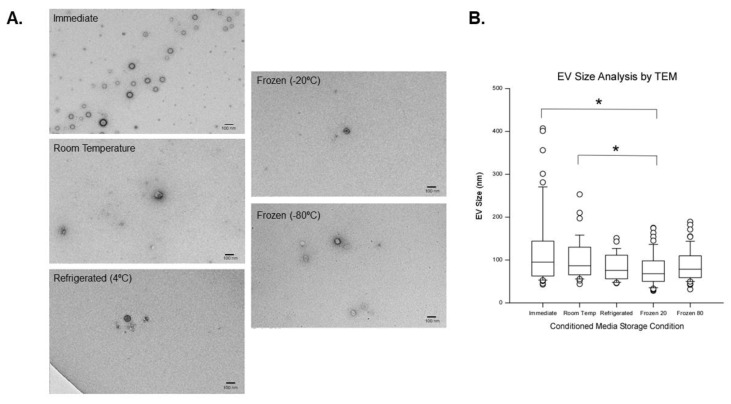
Transmission electron microscopy (TEM) analysis of extracellular vesicles (EVs). (**A**) TEM micrographs depicting EVs from all experimental groups that are roughly spherical ranging in diameter from 67.9–95.0 nm. Note that the morphology of EVs in the immediate group are similar to those in the −80 °C group with the distinct black ring (doughnut shape) indicating a bilayer structure. The calibration bar in micrographs is 100 nm. (**B**) TEM estimate of EV size. Significant differences were observed between EVs in the immediate group and −20 °C (*p* = 0.002) and between EVs in the room temperature group and −20 °C (*p* = 0.045). Data are presented as the median, 25th, and the 75th percentile in the box and whiskers show the 10th and 90th percentile. Potential outliers are depicted as open circles. Asterisk depicts significant difference (*p*-value < 0.05).

**Figure 6 ijms-23-07716-f006:**
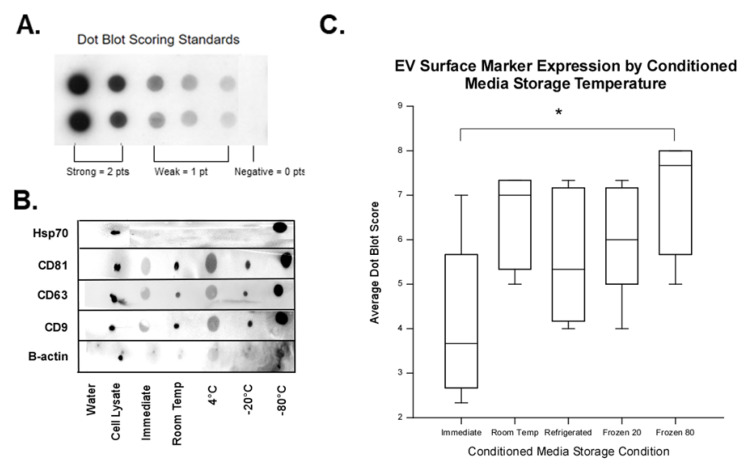
Effect of CM storage condition on dot blot staining. (**A**) Dot blot scoring standard provided to independent, blinded scorers to assess the intensity of EV characterization marker staining. Strong positive staining was assigned 2 points, weak positive staining was assigned 1 point, and negative staining 0 points. (**B**) Representative dot blots showing the expression of tetraspanins: clusters of differentiation (CD)9, CD63, and CD81; heat shock protein (Hsp) 70; and protein loading control beta actin (β-actin) by (left to right): water (negative control), whole MSC lysate (positive control), EVs isolated from CM of MSCs at storage conditions (immediate, room temperature, 4 °C, −20 °C and −80 °C). Note that while equal amounts of protein were blotted on the membrane, the −80 °C storage condition had the strongest apparent staining. (**C**) Dot blot intensity scores were averaged from three blinded, independent scorers and summed for each MSC line in all experimental conditions for analysis. Dots were scored on an ordinal integer scale depicted in (**A**). Maximum score (strong positive for all markers) was 8.0. Expression of β-actin was not scored as it was a positive control marker. Significant differences were detected in dot blot scores of EVs from the immediate group compared to −80 °C group (*p* = 0.005). This indicates that EVs from CM stored at −80 °C has stronger expression of characteristic EV surface markers than EVs isolated immediately from CM. Data are presented as the median, 25th, and the 75th percentile in the box and whiskers show the 10th and 90th percentile. Potential outliers are depicted as open circles. Asterisk depicts significance (*p*-value < 0.05), *n* = 5.

**Figure 7 ijms-23-07716-f007:**
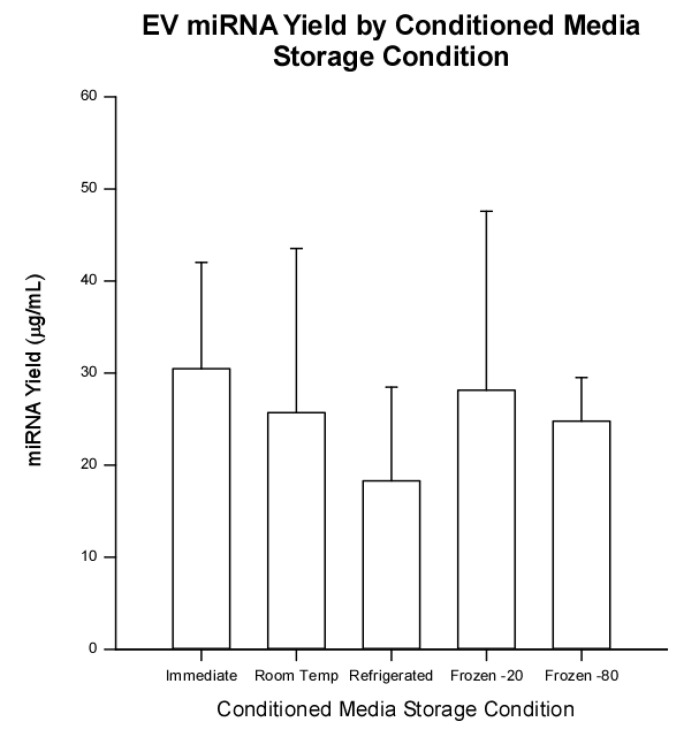
Total microRNA yield (µg/mL) from extracellular vesicles isolated from human umbilical cord-derived mesenchymal stromal cell conditioned media immediately or stored at room temperature, 4 °C, −20 °C, or −80 °C. The trend was for RNA from CM stored at 4 °C to have the smallest yield (18.3 µg/mL). Yields from immediate (30.5 µg/mL), room temperature (25.7 µg/mL), −20 °C (28.1 µg/mL), and −80 °C (24.8 µg/mL) were consistent. No significant differences were noted (*p* = 0.07). Data are presented as mean ± standard deviation, *n* = 5.

**Figure 9 ijms-23-07716-f009:**
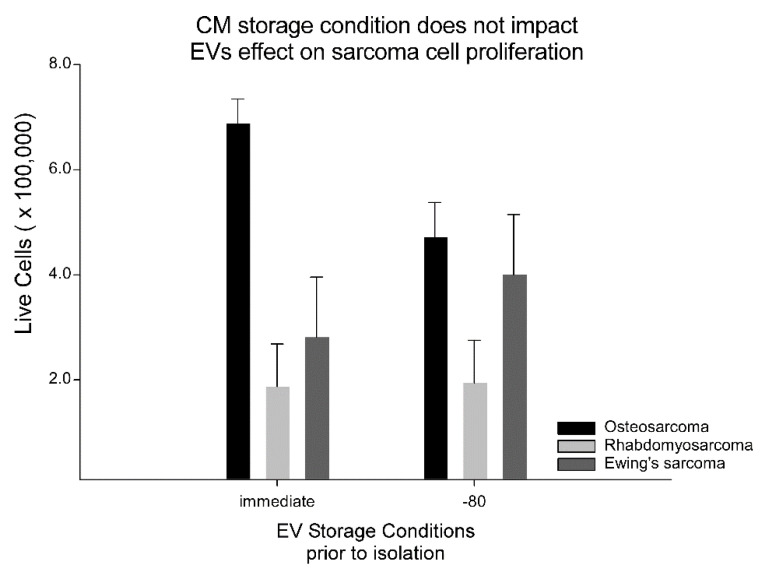
Storage condition prior to EV isolation had no effect on sarcoma cell proliferation. ANOVA main effects (STORAGE CONDTION or DOSE), and interactions terms (DOSE × STORAGE CONDITION) and (DOSE × STORAGE CONDITION × CANCER TYPE) indicated there was insufficient evidence to reject the null hypothesis. Data graphed here are the composite means from the ANOVA table (STORAGE CONDITION × CANCER TYPE and the associated standard error of the mean, SEM). Thus, there was no effect of EV dose (0, 5 × 10^5^, 1 × 10^6^, 1 × 10^7^, 1 × 10^8^ EVs per well), sarcoma cell line tested (seven lines), or pre-processing storage condition on sarcoma cell proliferation. Note 4 different osteosarcoma cell lines, 1 Ewing’s sarcoma cell line, and 2 rhabdomyosarcoma cell lines were tested.

**Table 1 ijms-23-07716-t001:** NTA Size of EVs by Storage Condition.

CM Storage Condition	NTA Size Mode (nm) ± SD
Immediate	85.08 ± 21.7
Room Temperature	96.16 ± 19.7
4 °C	88.48 ± 8.1
−20 °C	105.42 ± 18.6
−80 °C	100.40 ± 21.9 N = 5 per storage condition

**Table 2 ijms-23-07716-t002:** EVs released per cell by line.

Cell Line	EVs Released per Cell (Mean ± SD)
HUC 284	1.16 × 10^4^ ± 1.06 × 10^4^
HUC 293	2.29 × 10^4^ ± 9.55 × 10^3^
HUC 297	1.51 × 10^4^ ± 3.28 × 10^3^
HUC 298	3.20 × 10^4^ ± 2.07 × 10^4^
HUC 301	2.78 × 10^4^ ± 9.88 × 10^3^

Note: EV release per cell was averaged across the 5 storage conditions.

**Table 3 ijms-23-07716-t003:** RT-PCR Primers.

miRNA Primer	Sequence (5′ to 3′)	Tm (°C)	Accession Number
HU.MIR7641	GCAGTTGATCTCGGAAGCT	54.9	MIMAT0029782
	GTCCAGTTTTTTTTTTTTTTTGCTTAG	51.6	
HU.MIR6089	ACAGGAGGCCGGGGTGGG	65.7	MIMAT0023714
	TCCAGTTTTTTTTTTTTTTTCCGC	52.2	
HU.MIR4792	ACAGCGGTGAGCGCTCG	61.3	MIMAT0019964
	GTCCAGTTTTTTTTTTTTTTTGCCAG	53.4	
HU.MIR4466	ACAGGGGTGCGGGCCGG	67.4	MIMAT0018993
	TCCAGTTTTTTTTTTTTTTTCCCCG	53.8	
HU.MIR3665	CAGAGCAGGTGCGGGGC	62.5	MIMAT0018087
	TCCAGTTTTTTTTTTTTTTTCGCC	52.2	
HU.MIR1273e	CAGTTGCTTGAACCCAGGAA	55.5	MIMAT0018079
	GTCCAGTTTTTTTTTTTTTTTCCAC	51.0	
HU.MIR1246	GCAGAATGGATTTTTGGAG	49.2	MIMAT0005898
	GTCCAGTTTTTTTTTTTTTTTCCTG	50.8	
HU.MIR658	GGCGGAGGGAAGTAGGTC	57.4	MIMAT0003336
	GTCCAGTTTTTTTTTTTTTTTACCAAC	51.6	
HU.MIR127	AGTCGGATCCGTCTGAGCTT	58.0	MIMAT0000446
	GTCCAGTTTTTTTTTTTTTTTAGCC	50.9	
MystiCq Universal PCR Primer	Complementary to adapter sequence added during cDNA synthesis		Cat. No. MIRUP

**Table 4 ijms-23-07716-t004:** Calculation of Viable Cell Number.

Cell Line	Passage	Previous Passage PDT (Hours)	Time in Culture (Hours)	Cells Seeded	Population Doublings	Estimated Viable Cell Number
HUC 284	P5	58.7	72	150,000	1.23	3.34 × 10^6^
HUC 293	P5	59.0	72	150,000	1.22	3.33 × 10^6^
HUC 297	P6	38.4	96	150,000	2.50	5.25 × 10^6^
HUC 298	P4	46.2	96	150,000	2.08	4.62 × 10^6^
HUC 301	P6	51.6	72	150,000	1.39	3.59 × 10^6^

## Data Availability

The original datasets generated here are available following completion of material transfer agreement.

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
