# Peer review of "Effect of Pre-Processing Storage Condition of Cell Culture-Conditioned Medium on Extracellular Vesicles Derived from Human Umbilical Cord-Derived Mesenchymal Stromal Cells"

_ijms, 2022, doi:10.3390/ijms23147716_

Round 1
Reviewer 1 Report
In this manuscript, Wright et al. assessed the effect of storage temperatures of conditioned medium on extracellular vesicles derived from derived from HUC-MSCs. The topic is interesting and to a large extent I agree the discovery here is important. However, I have some concerns that need to be addressed:
1. How the conditioned medium was collected for the different storage conditions?
2. All the assessment were done after isolation. The isolation method used for this study is size exclusion chromatography. Better to try at least another method to see whether it matters? Is it possible to check the effect on extracellular vesicles directly in the CM?
3. For figure 9, where to find the data of different EV dose? A positive control is needed to get useful information.
Author Response
Changes to EV Pre-processing Storage Condition manuscript in response to Reviewer’s comments
Reviewer 1
In this manuscript, Wright et al. assessed the effect of storage temperatures of conditioned medium on extracellular vesicles derived from derived from HUC-MSCs. The topic is interesting and to a large extent I agree the discovery here is important. However, I have some concerns that need to be addressed:
- How the conditioned medium was collected for the different storage conditions?
The reviewer indicated that the methods for collection of conditioned medium needed further clarification. To address this comment, the methods section was edited (see comment labeled Reviewer #1, comment 1 in manuscript).
- All the assessment were done after isolation. The isolation method used for this study is size exclusion chromatography. Better to try at least another method to see whether it matters? Is it possible to check the effect on extracellular vesicles directly in the CM?
The reviewer is correct that more information would be provided by showing whether the effect of pre-processing storage condition affects other isolation methods. We selected the SEC isolation methods since SEC is a popular alternative to ultracentrifugation for EV isolation, and since SEC is more easily scalable. Since it was not feasible to repeat the experiment using a different isolation method, we address this comment by adding a couple of sentences to the discussion pointing out this limitation of our work (see comment labeled Reviewer #1, comment 2 in manuscript).
- For figure 9, where to find the data of different EV dose? A positive control is needed to get useful information.
The reviewer has concerns about the data presented in figure 9. To address these concerns, we added information indicating that the data in the graph came from the ANOVA table (STORAGE CONDITION x CANCER TYPE group means and standard error of the mean, SEM) to the caption of figure 9 (see comment labeled Reviewer #1, comment 3 in the manuscript).
- We thank Reviewer #1 for their constructive comments and believe that we have addressed all of them. Their improvements were incorporated in the revised manuscript.
Reviewer 2 Report
Wright et al. and colleagues studied the Effect of pre-processing storage condition of cell culture conditioned medium on extracellular vesicles derived from human umbilical cord-derived mesenchymal stromal cells. This study is interesting and well-written.
Specific Comments.
The introduction, materials and method are well written.
The author must do the following EVs/Exosome markers such as CD9, CD63, CD9 and ALIX etc in all conditions by western blots, as dots blot not well represent the size of markers.
The dot blots quality is poor, and the blot seems fragmented
Figure 4 legend is missing
The p values (*) in missing in Figure 4A-B.
The discussion is well discussed with previous papers.
Author Response
Reviewer 2
Wright et al. and colleagues studied the Effect of pre-processing storage condition of cell culture conditioned medium on extracellular vesicles derived from human umbilical cord-derived mesenchymal stromal cells. This study is interesting and well-written.
Specific Comments.
The introduction, materials and method are well written.
- We thank Reviewer #2 for this positive comment.
The author must do the following EVs/Exosome markers such as CD9, CD63, CD9 and ALIX etc in all conditions by western blots, as dots blot not well represent the size of markers.
The reviewer indicates that EV specific surface markers should be evaluated by western blots and that dot blot does not represent the size of the proteins. We agree with the reviewer that western blot does provide more information and more specific information about the blotted proteins. Unfortunately, the characterization of EVs by western blot consumes a good deal of the sample. The EVs isolated by SEC had low protein concentration and the only way we could obtain consistent staining was by lipolyzing the samples and reconstituting them in a small volume and dotting them. To address the reviewer’s concern, we added to the Discussion pointing out this limitation of our work (see comment labeled Reviewer #2, comment 1 in manuscript).
The dot blots quality is poor, and the blot seems fragmented
The reviewer indicates that the quality of the dot blots is poor. We agree with the reviewer that the chemiluminescent scanner did a poor job recording the data and they look blotchy. Unfortunately, we cannot recover the chemiluminescence and rescan. While the dot blots did not produce the prettiest data, they used considerably less EV sample compared to western blots. Commentary to this effect was added to Discussion (cited above). WE ARE UNABLE TO RESCAN THE BLOTS. WE CANNOT ADDRESS THE ISSUE RAISED BY THE REVIEWER. NO CHANGES TO MANUSCRIPT.
Figure 4 legend is missing
The reviewer is correct. Thank you for catching this omission. To address the reviewer’s concern, the caption was added (see comment labeled Reviewer #2, comment 2 in manuscript).
The p values (*) in missing in Figure 4A-B.
The reviewer indicates that asterisks indicating significant differences is missing in Figure 4A-B. As can be seen in Figure 4A, an asterisk indicating significant differences in polydispersity index between the immediately processed EV samples and those processed after storage at -80ºC is shown. However, there were no significant effects of pre-processing storage condition on zeta potential or dynamic light scattering based particle size. This is indicated in the text, and thus, no asterisk is found in Figure 4B or 4C. NO CHANGES TO MANSCRIPT.
The discussion is well discussed with previous papers.
- We thank Reviewer #2 for this positive comment.
- We thank Reviewer #2’s for their constructive comments, and while we could not address all of them, we did what we could do. Their improvements were incorporated in the revised manuscript.
Round 2
Reviewer 1 Report
The revised version is improved. However, I still have one question regarding the conditioned medium:
Based on my understanding, the CM was collected without centrifugation/filtration, and very likely it will contain cells/cell debris. Will these cells/cell debris affect the extracellular vesicles during storage?
Author Response
REVISION ROUND #2: Changes to EV Pre-processing Storage Condition manuscript in response to Reviewer’s comments
Reviewer 1
Comments and Suggestions for Authors
The revised version is improved.
We thank the Reviewer for acknowledging that the modification suggested in revision round 1 made an improvement to the Manuscript.
However, I still have one question regarding the conditioned medium:
Based on my understanding, the CM was collected without centrifugation/filtration, and very likely it will contain cells/cell debris. Will these cells/cell debris affect the extracellular vesicles during storage?
The reviewer expressed a concern that the conditioned medium (CM) would be contaminated by cells/ cell debris. We agree with the reviewer that without removal of cells/cellular debris the EV sample could be contaminated. Since the CM is clarified first by centrifugation and second by filtration, as indicated in the Manuscript, the reviewer’s concern, was addressed. We highlight this portion of the Ms by adding a comment labeled Revision round #2, Reviewer #1 comment in section 2.3 of the manuscript on page 4. NO CHANGES TO MANUSCRIPT.
Submission Date 09 June 2022
Date of this review 03 Jul 2022 00:02:08